# PCRNet: Phase-aware Complex Refinement Network for EEG-based Auditory Attention Decoding

Xiran Chen [† 1]   Xiaoke Yang [† 1]   Jian Zhou [1]   Zhao Lv [1]   Cunhang Fan [* 1]

## Abstract

Auditory attention decoding (AAD) based on Electroencephalography (EEG) aims to identify the attended speaker in multi-speaker environments. However, existing methods typically overlook the crucial phase information of EEG signals, which limits their ability to distinguish structured neural patterns from random noise in the frequency domain and hinders robust decoding. To address this issue, this paper proposes a Phase-aware Complex Refinement Network (PCRNet) for AAD, which consists of a Temporal Context Calibration (TCC) module and a Dual-Domain Integration (DDI) module. Specifically, the TCC module captures long-range temporal dependencies through multi-scale temporal attention mechanism, while the DDI module employs a phase-guided spectral filtering strategy to dynamically suppress noise-dominated frequencies and refine the real and imaginary components separately. This design enables effective phase recalibration and enhances the discriminability of target features in the complex domain. Experimental results on three public datasets demonstrate that PCRNet outperforms state-of-the-art methods, particularly under challenging ultra-short 0.1-second windows. Code is available at: https://github.com/SunshineGreeny/PCRNet.

## 1. Introduction

At a cocktail party (Haykin & Chen, 2005), humans can selectively focus on one speaker among multi-speaker environments, which motivates research on auditory attention decoding (AAD) that aims to identify the attended source from brain signals. According to previous neuroscience studies, there exists a correlation between brain activity and auditory attention (Mesgarani & Chang, 2012). Following these concepts, AAD could potentially enhance the design of human-centered intelligent interaction systems, such as hearing aids (O'sullivan et al., 2015; Geirnaert et al., 2021).

Neuroscientific evidence shows that auditory attention modulates cortical responses to stimuli through top-down mechanisms (Mesgarani & Chang, 2012; O'sullivan et al., 2015). Electroencephalography (EEG) signals are widely used due to their characteristics of non-invasive nature, ease of acquisition, and high temporal resolution (Niedermeyer & da Silva, 2005; Wolpaw, 2007). Consequently, this paper concentrates on utilizing EEG signals for AAD. However, EEG signals suffer from a low signal-to-noise ratio (SNR) and non-stationarity, primarily due to physiological artifacts and volume conduction (Urigüen & Garcia-Zapirain, 2015), which severely compromise signal quality. Crucially, neuroscientific studies have established that the phase of low-frequency cortical oscillations is the primary carrier of information for tracking dynamic speech streams (Luo & Poeppel, 2007; Obleser & Kayser, 2019) and encoding critical temporal structural information (Oppenheim & Lim, 2005). Therefore, effectively recovering phase information from noisy backgrounds remains a critical challenge for AAD.

Prior work has focused primarily on the temporal and spatiotemporal dependencies, utilizing architectures like Long Short-Term Memory (LSTM) networks or attention mechanisms to model raw neural dynamics (Monesi et al., 2020; Su et al., 2022; Vaswani et al., 2017; Jiang et al., 2022). However, considering the time-varying or spectral-spatial information in EEG signals, EEG features cannot be comprehensively represented using conventional methods. Building on this, later methods have integrated spectral information, developing hybrid temporal-frequency or spatial-frequency frameworks to leverage the distinct power distributions across frequency bands (Cai et al., 2021; Ni et al., 2024; Zhu et al., 2025). Classical approaches utilize filter bank common spatial patterns to extract discriminative energy

†These authors contributed equally. * Corresponding author. [1] State Key Laboratory of Opto-Electronic Information Acquisition and Protection Technology, (School of Computer Science and Technology), Anhui University, Hefei, 230601, Anhui, P.R. China . Correspondence to: Cunhang Fan <cunhang.fan@ahu.edu.cn>.

*Proceedings of the 43rd International Conference on Machine Learning*, Seoul, South Korea. PMLR 306, 2026. Copyright 2026 by the author(s).

variance from specific sub-bands (e.g., Theta and Alpha), which are critical for attention modulation (Ang et al., 2008; Chin et al., 2009). In the deep learning era, methods often convert raw signals into power spectral density maps or Differential Entropy features, utilizing Convolutional Neural Networks (CNNs) or Graph Neural Networks (GNNs) to model regional energy fluctuations (Zheng & Lu, 2015; Song et al., 2018; Zhong et al., 2020; Fan et al., 2024a). Furthermore, strictly treating temporal-frequency spectrograms as 2D images allows for the application of computer vision backbones to capture texture-like patterns in the magnitude domain (Tabar & Halici, 2016; Bashivan et al., 2015). To process this information, researchers have developed dual-branch (Ni et al., 2024; Yan et al., 2024) or multi-view frameworks (Li et al., 2025; Fan et al., 2025a) that simultaneously model the temporal evolution and the spectral energy variation, effectively leveraging the power differences to distinguish attentional states (Zhu et al., 2025). However, despite these multi-domain advancements, these methods all predominantly rely on amplitude or power, treating frequency representations merely as intensity maps but face the critical challenge of missing phase information. Neuroscientific evidence indicates that the brain tracks dynamic speech streams primarily through phase-locking mechanisms (neural entrainment) rather than mere power modulation (Golumbic et al., 2013; Obleser & Kayser, 2019). It limits their capacity to differentiate structured neural patterns from random noise in the frequency domain, especially in low-SNR conditions.

To address these issues, this paper proposes a phase-aware complex refinement network for AAD, named PCRNet, which effectively synergizes temporal-frequency complementary features and captures global spectral patterns to distinguish structured neural patterns from random noise. Specifically, our model consists of two key modules: (1) *Temporal Context Calibration (TCC) Module*. In this module a multi-scale self-attention mechanism with a gating mechanism is applied to ensure independent temporal calibration for robust feature refinement prior to frequency domain analysis. (2) *Dual-Domain Integration (DDI) Module*. This module features a novel residual-spectral interface (RSI) block, which leverages an amplitude importance gate and complex feature mixer to suppress spectral noise and fuse complementary real and imaginary frequency information, effectively achieving effective phase recalibration and the discriminability of target features. Finally, the refined features are passed to a spatial layer and a temporal layer, followed by a global average pooling applied to obtain richer representations. The major contributions of this paper are summarized as follows:

- The proposed PCRNet establishes a task-driven phase-aware dual-domain paradigm for AAD. This architecture integrates local temporal dynamics with global spectral representations to capture comprehensive neu-

rophysiological features from EEG signals.
- A novel DDI module is designed to refine decoupled real and imaginary spectral information and enhance the discriminability of target representations in the complex domain by employing phase-guided filtering.
- PCRNet achieves state-of-the-art (SOTA) decoding accuracy across three public datasets and demonstrates superior robustness, particularly under challenging ultra-short 0.1-second decision windows.

## 2. The Proposed PCRNet Method

Previous AAD methods overlook the crucial phase information in EEG signals. To address these issues, we propose PCRNet, which mainly consists of TCC and DDI modules. Figure 1 illustrates the overall structure of PCRNet. It effectively captures long-range temporal dependencies and suppresses noise-dominated frequencies.

Given the EEG data split by a moving window, a series of decision windows is obtained, each containing a short segment of EEG signals. Consider the original EEG data of a decision window represented by $E = [e_1, \cdots, e_i, \cdots, e_T] \in \mathbb{R}^{C \times T}$, where C is the number of EEG channels and T is the length of the decision window. Here, $e_i \in \mathbb{R}^{C \times 1}$ represents the EEG data at the $i$-th time point within $E$. We aim to learn a representation $G(\cdot)$, which maps the EEG input $E$ to the corresponding label $y = G(E)$. Here, $y$ denotes the locus of auditory attention (i.e., left or right). Before inputting the EEG data into PCRNet, a common spatial patterns (CSP) algorithm (Blankertz et al., 2007; Ramoser et al., 2000) is employed to extract raw features from the EEG data under different brain states. Following these processes $\tilde{E} \in \mathbb{R}^{C \times T}$ is obtained.

### 2.1. Temporal Context Calibration Module

Standard channel attention mechanisms often compute correlations directly on raw features, potentially ignoring significant temporal variations. To address these limitations, multi-scale temporal attention (MTA) (Li et al., 2025) has been proposed. Inspired by this method, we proposed a multi-scale self-attention mechanism along with a gating mechanism to comprehensively capture multi-scale spatiotemporal representations from EEG signals and ensure more robust and fine-grained feature refinement before global integration.

Firstly, we split the tensor $\tilde{E}$ along the channel dimension to obtain the $Q$, $K$ and $V$ components proposed by the self-attention mechanism (Vaswani et al., 2017). This can be formulated as follows:

$$Q, K, V = Split(\tilde{E}) \in \mathbb{R}^{C \times 1 \times T} \tag{1}$$

where $Split(\cdot)$ denotes the operation of splitting $\tilde{E}$ along

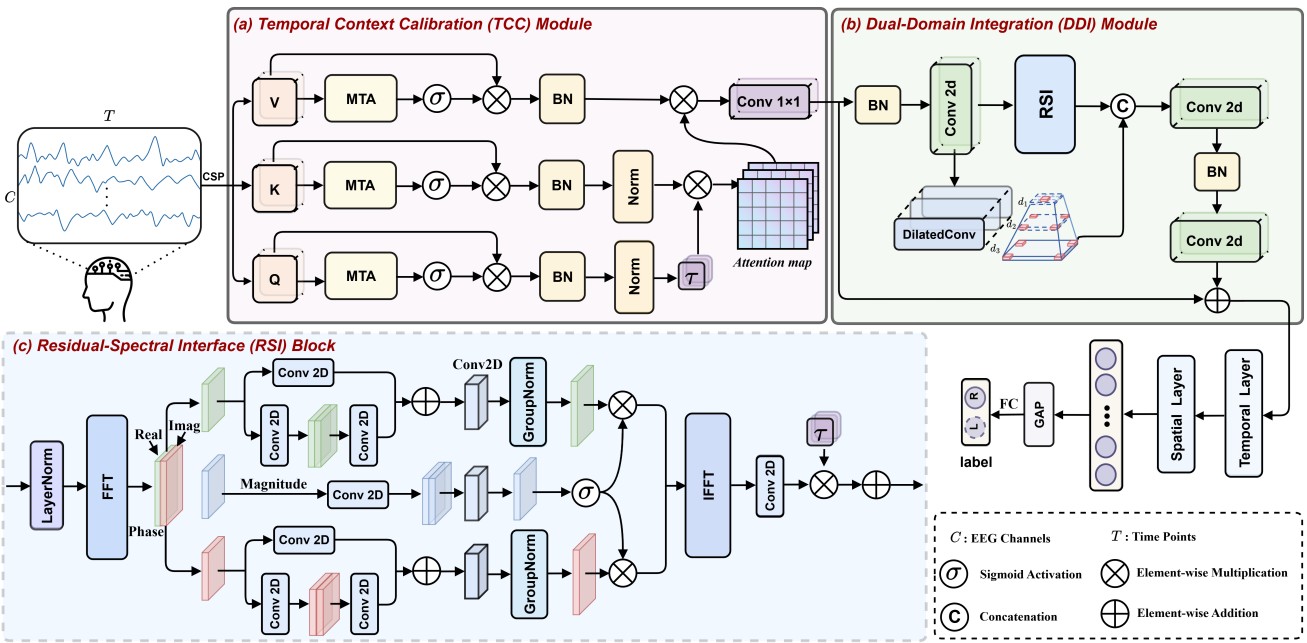

*Figure 1.* The overall architecture of our PCRNet model for AAD, which mainly consists of two modules: (a) temporal context calibration (TCC) module and (b) dual-domain integration (DDI) module, where $d_i$ ($i \in \{1, 2, 3\}$) represents the kernel size used in the dilated concolution. The outputs are two predicted labels (right/left) related to auditory attention.

the channel dimension.

Then we utilize a multi-scale self-attention mechanism to refine the $Q$, $K$ and $V$ projections and construct expressive features $Q'$, $K'$, $V' \in \mathbb{R}^{C \times 1 \times T}$ with richer imformation. The refinement of MTA effectively aggregates temporal dependencies over multiple receptive fields.

$$Q' = Q \odot (1 + \sigma(MTA(Q))) \qquad (2)$$

$$K' = K \odot (1 + \sigma(MTA(K))) \qquad (3)$$

$$V' = V \odot (1 + \sigma(MTA(V))) \qquad (4)$$

where $\odot$ denotes the element-wise multiplication operation, and $\sigma(\cdot)$ denotes the sigmoid activation operation. The residual term ensures that the refinement is scale-preserving and stabilizes optimization in subsequent attention computation.

After that, we proceed with the standard self-attention calculation. The features are reshaped and normalized to compute the channel-wise attention map. Meanwhile, we employ a $1 \times 1$ convolutional layer to obtain more robust spatiotemporal features $H$ for the next block. The process can be formalized as follows:

$$Attention(Q', K', V') = Softmax(\tau \cdot \frac{Q' \cdot K'^{T}}{\sqrt{d_k}}) \cdot V'$$
$$(5)$$

$$H = Conv(Attention(Q', K', V')) \in \mathbb{R}^{1 \times C \times T} \qquad (6)$$

where $\tau$ is a learnable temperature parameter and $d_k$ is the scaling factor.

Then the EEG data $H$ is fed into DDI for further processing, capturing and fusing complementary spectral information.

$$F = DDI(H) \in \mathbb{R}^{1 \times C \times T} \qquad (7)$$

The DDI will be elaborated in the following subsection.

### 2.2. Dual-Domain Integration Module

Previous studies have considered both time-varying characteristics and spectral-spatial information through temporal-frequency fusion (Ni et al., 2024). Inspired by this, the DDI module is designed to capture long-range temporal contexts and phase information by processing features in parallel temporal and spectral branches within a residual learning framework.

Firstly, $\mathbf{X}_{in}$ is processed by a projection block to map the calibrated features into a high-dimensional latent space suitable for dual-domain analysis. The projected feature $\mathbf{H}_{proj}$ is formulated as follows:

$$\mathbf{H}_{proj} = GELU(Conv(BN(Conv(\mathbf{X}_{in})))) \qquad (8)$$

where $GELU(\cdot)$ denotes a Gaussian error linear unit activation function, $Conv(\cdot)$ denotes a 1D convolution operation and $BN(\cdot)$ represents a batch normalization operation.

The projected features are then fed into the temporal branch to capture long-range dependencies. To expand the temporal receptive field without losing resolution, we employ a cascade of dilated convolutions with exponentially increasing dilation rates. Let $\mathcal{D}_{1 \times d}(\cdot)$ denote a dilated convolution

function and the dilation rate $d$ is set to $d \in \{d_1, d_2, d_3\}$. The hierarchical temporal features are extracted sequentially through the cascade of dilated convolutions, and the final aggregated temporal representation $\mathbf{H}_{time}$ is given as follows:

$$\mathbf{H}_{time} = \mathcal{D}_{1 \times d_3}(\mathcal{D}_{1 \times d_2}(\mathcal{D}_{1 \times d_1}(\mathbf{H}_{proj}))) \qquad (9)$$

Simultaneously, the parallel spectral branch processes $\mathbf{H}_{proj}$ using RSI to refine features in the frequency domain. The spectral feature is denoted as:

$$\mathbf{H}_{freq} = RSI(\mathbf{H}_{proj}) \qquad (10)$$

The RSI block will be detailed in the following subsection.

To synthesize the complementary information, the outputs from both branches are concatenated along the channel dimension and fused through a transition block $\mathcal{F}_{trans}(\cdot)$, which shares the same structure as the projection block. Finally, a global residual connection is applied to preserve the original signal information. The final output $F$ is computed as:

$$F = \mathbf{X}_{in} + \mathcal{F}_{trans}(Concat(\mathbf{H}_{time}, \mathbf{H}_{freq})) \qquad (11)$$

where $Concat(\cdot)$ denotes the concatenation operation, and the addition represents the element-wise residual connection.

### 2.3. Residual-Spectral Interface Block

Considering that existing spectral feature extraction methods often overlook the phase components, which have been shown to contain essential structural information for neural decoding (Oppenheim & Lim, 2005; Golumbic et al., 2013), we introduce RSI, a phase-guided spectral refinement method, to dynamically suppress noise-dominated frequencies and effectively recalibrate phase.

Given an input feature map $X \in \mathbb{R}^{C \times H \times W}$, we first apply layer normalization to standardize the feature distribution. Next, we transform the features into the frequency domain. The process can be formally described as follows:

$$Z = \mathcal{F}(LN(X)) \qquad (12)$$

where $Z$ contains both real ($R$) and imaginary ($I$) components. $\mathcal{F}$ denotes the 2D Fast Fourier Transform and $LN(\cdot)$ denotes a layer normalization operation.

To adaptively suppress noise and highlight informative frequency bands, we introduce an amplitude importance gate. We compute the amplitude spectrum $A$ and pass it through a channel-wise gating network to generate a spectral mask $M$. The process can be computed as:

$$A = |Z| = \sqrt{Z_{real}^2 + Z_{imag}^2} \qquad (13)$$

$$M = \sigma(Conv(\delta(Conv(A)))) \qquad (14)$$

where $Conv(\cdot)$ denotes a $1 \times 1$ convolution layer, $\delta$ is the Leaky ReLU activation operation, and $\sigma$ is the sigmoid function.

To preserve phase information while refining the spectral representation, we employ a complex feature mixer $\Psi(\cdot)$. The real and imaginary parts are processed independently by a dual-branch convolutional structure and then modulated by the amplitude mask $M$. This process is formulated as follows:

$$\tilde{\mathbf{R}} = \Psi_{real}(R) \odot M \qquad (15)$$

$$\tilde{\mathbf{I}} = \Psi_{imag}(I) \odot M \qquad (16)$$

where $\odot$ denotes element-wise multiplication. The mixing function $\Psi(\cdot)$ is defined as:

$$\Psi(x) = \mathrm{GN}(\mathbf{W}_{out} \cdot (\mathcal{G}(x) + \mathbf{W}_{res} \cdot x)) \qquad (17)$$

where $GN$ represents the group normalization operation. $W_{res}$ and $W_{out}$ denote the learnable weight matrices of the residual projection layer and the output projection layer, respectively. The non-linear branch $\mathcal{G}(\cdot)$ is defined as:

$$\mathcal{G}(x) = \mathbf{W}_2 \cdot GELU(\mathbf{W}_1 \cdot x) \qquad (18)$$

where $\mathbf{W}_2$ and $\mathbf{W}_1$ are the learnable weights of the two convolutional layers, and $GELU(\cdot)$ denotes a Gaussian error linear unit activation function.

The refined components are recombined into a complex tensor $\tilde{\mathbf{Z}}$. We reconstruct the spatial-temporal features using the Inverse Fast Fourier Transform (FFT), followed by a post-processing projection. It can be formulated as follows:

$$\tilde{\mathbf{Z}} = \tilde{\mathbf{R}} + j \cdot \tilde{\mathbf{I}} \qquad (19)$$

$$F_{spectral} = GELU(W_{post} \cdot \mathcal{F}^{-1}(\tilde{\mathbf{Z}}) + b_{post}) \qquad (20)$$

where $W_{post}$ and $b_{post}$ denote the learnable weights and bias of the convolutional layer, respectively, which projects the globally filtered features into the target channel space. $\mathcal{F}^{-1}(\cdot)$ represents the inverse FFT.

Finally, we adopt layer scaling to facilitate gradient flow and stabilize training in deep networks. We introduce a learnable diagonal matrix $\mathbf{H}_{freq}$ (initialized to zero) to weight the spectral branch. The process can be formalized as follows:

$$\mathbf{H}_{freq} = X + \gamma \cdot F_{spectral} \qquad (21)$$

where $\gamma$ represents a learnable channel-wise scaling factor initialized to zero.

## 2.4. Feature extraction

Finally, specific convolutional layers, including a temporal layer and a spatial layer, are applied to aggregate the extracted spatiotemporal information, followed by an adaptive average pooling layer for feature downsampling. The resulting representation is then fed into a fully connected layer to produce the final output $p$. This process is expressed as follows:

$$F' = GELU(BN(TemporalConv(F))) \quad (22)$$

$$F'' = GELU(BN(SpatialConv(F'))) \quad (23)$$

$$O = AdaptiveAvgPool(F'') \quad (24)$$

$$p = FC(Flatten(O)) \quad (25)$$

where $TemporalConv(\cdot)$ denotes a 2D convolutional layer with a $1 \times 2$ kernel size, and $SpatialConv(\cdot)$ denotes a 2D convolutional layer with a $C \times 1$ kernel size across all EEG channels. $BN(\cdot)$ denotes batch normalization, and $AdaptiveAvgPool(\cdot)$ denotes an adaptive average pooling layer. Flatten$(\cdot)$ reshapes the extracted feature maps into a one-dimensional vector, while FC$(\cdot)$ denotes a fully connected layer that maps the flattened features to the label space for classification.

## 3. Experiments

### 3.1. Datasets

In this paper, we conduct experiments on three publicly available datasets, namely KUL (Das et al., 2016; 2019), DTU (Fuglsang et al., 2017; 2018), and AVED (Fan et al., 2024b). KUL and DTU are commonly used datasets that contain EEG data only from audio-only scenarios. AVED contains EEG data from both audio-only and audio-visual scenarios(Fan et al., 2025b). The details of these datasets are listed in Table 1.

1) **KUL Dataset**: In this dataset, 64-channel EEG data were collected from 16 normal-hearing subjects (8 males and 8 females) at a sampling rate of 8192 Hz. The stimuli contained four Dutch short stories narrated by three male Flemish speakers. Each subject completed 8 trials, which lasted 6 minutes.
2) **DTU Dataset**: In this dataset, 64-channel EEG data were collected from 18 normal-hearing subjects at a sampling rate of 512 Hz. The stimuli contained Danish audiobooks narrated by a male and a female speaker. Each subject completed 60 trials, which lasted 50 seconds.
3) **AVED Dataset**: In this dataset, 32-channel EEG data were collected from 20 normal-hearing subjects (14 males and 6 females) at a sampling rate of 1 kHz. The subjects were divided into two groups of 10. One underwent audio-only experiments, and the other underwent audio-visual ones. All the stimuli contained 16 Chinese

short stories narrated by a male and a female speaker. Each subject completed 16 trials, which lasted 152 seconds.

### 3.2. Data Processing

To ensure a fair comparison of PCRNet with existing methods, specific preprocessing steps are applied to each dataset. For the KUL dataset, the EEG data are first re-referenced to the average response of the mastoid electrodes and then band-pass filtered between 0.1 Hz and 50 Hz. Finally, the data are downsampled to 128 Hz. For the DTU dataset, the EEG data are filtered to remove 50 Hz line noise and its harmonics. Eye artifacts are removed through joint decorrelation, and the EEG data are re-referenced to the average response of the mastoid electrodes. Finally, the EEG data are downsampled to 64 Hz. For the AVED dataset, a notch filter is first applied to remove power-line interference at 50 Hz. Next, a finite impulse response filter is used for high-pass and low-pass filtering to suppress noise. Then, the EEG data are downsampled to 128 Hz, followed by independent component analysis for further artifact removal. Finally, all EEG channels are re-referenced to ensure consistency and comparability.

*Table 1.* Details of three datasets used in the experiments.

| Dataset | Subjects | Scene | Channels | Duration (minutes) | Language |
|---------|----------|-------|----------|--------------------|----------|
| KUL | 16 | audio-only | 64 | 48 | Dutch |
| DTU | 18 | audio-only | 64 | 50 | Danish |
| AVED | 10 | audio-only | 32 | 40 | Mandarin |
| | 10 | audio-visual | 32 | 40 | Mandarin |

### 3.3. Implementation Details

In previous AAD studies, the accuracy of the auditory attention classification has been used as a baseline for the evaluation of model performance. We follow this convention and evaluate our proposed PCRNet on the KUL, DTU, and AVED datasets. All experiments are conducted using PyTorch on an RTX 4090 GPU. In the following, we take the KUL dataset with a 1-second decision window as an example to illustrate implementation details, including training settings and network configuration.

First, we set the proportions of the training, validation, and test sets to 8:1:1. For each subject in the KUL dataset, this split yields 4,600 decision windows for training, 576 for validation, and 576 for testing. Meanwhile, we set the batch size to 32 and the maximum number of epochs to 100, and employ an early stopping strategy to mitigate the risk of overfitting and improve the model's generalization ability. This mechanism monitors the validation loss at the end of each epoch. If the validation loss does not decrease for 10

*Table 2.* Auditory attention detection accuracy (%) comparison on the KUL, DTU and AVED datasets. The results annotated by $*$ for AVED are reproduced, and the remaining results replicated from the corresponding paper. Best results are highlighted in bold.

| Decision Window | Model | Dataset | | | |
|---|---|---|---|---|---|
| | | KUL | DTU | AVED (AO) | AVED (AV) |
| 0.1-second | SSF-CNN (Cai et al., 2021) | $76.3 \pm 8.47$ | $62.5 \pm 3.40$ | $53.3 \pm 1.91$ | $54.2 \pm 2.00$ |
| | MBSSFCC (Jiang et al., 2022) | $79.0 \pm 7.34$ | $66.9 \pm 5.00$ | $57.6 \pm 2.87$ | $58.9 \pm 2.60$ |
| | DBPNet (Ni et al., 2024) | $85.3 \pm 6.22$ | $74.0 \pm 5.20$ | $53.6 \pm 2.93$ | $55.7 \pm 2.45$ |
| | DARNet (Yan et al., 2024) | $89.2 \pm 5.50$ | $74.6 \pm 6.09$ | $51.3 \pm 3.50$ | $50.3 \pm 0.60$ |
| | SSF-DST$^*$ (Zhu et al., 2025) | $92.1 \pm 5.70$ | $79.7 \pm 6.07$ | $53.1 \pm 2.89$ | $54.9 \pm 3.84$ |
| | MHANet (Li et al., 2025) | $95.6 \pm 4.83$ | $75.5 \pm 5.68$ | $67.9 \pm 2.10$ | $67.4 \pm 3.24$ |
| | **PCRNet (ours)** | $\mathbf{98.4 \pm 2.83}$ | $\mathbf{78.6 \pm 7.77}$ | $\mathbf{70.2 \pm 1.72}$ | $\mathbf{69.2 \pm 2.65}$ |
| 1-second | SSF-CNN (Cai et al., 2021) | $84.4 \pm 8.67$ | $69.8 \pm 5.12$ | $57.1 \pm 3.54$ | $59.2 \pm 5.13$ |
| | MBSSFCC (Jiang et al., 2022) | $86.5 \pm 7.16$ | $75.6 \pm 6.55$ | $70.5 \pm 3.92$ | $69.5 \pm 5.77$ |
| | DBPNet (Ni et al., 2024) | $94.4 \pm 4.62$ | $79.8 \pm 6.91$ | $58.7 \pm 3.60$ | $62.0 \pm 4.92$ |
| | DARNet (Yan et al., 2024) | $94.8 \pm 4.53$ | $80.1 \pm 6.85$ | $80.6 \pm 15.69$ | $83.1 \pm 11.64$ |
| | SSF-DST$^*$ (Zhu et al., 2025) | $96.0 \pm 4.48$ | $84.1 \pm 6.15$ | $70.3 \pm 4.49$ | $69.3 \pm 5.49$ |
| | MHANet (Li et al., 2025) | $95.8 \pm 4.29$ | $82.2 \pm 8.13$ | $87.1 \pm 4.48$ | $86.0 \pm 5.32$ |
| | **PCRNet (ours)** | $\mathbf{98.5 \pm 2.35}$ | $\mathbf{85.3 \pm 7.27}$ | $\mathbf{89.7 \pm 3.49}$ | $\mathbf{89.5 \pm 3.16}$ |
| 2-second | SSF-CNN (Cai et al., 2021) | $87.8 \pm 7.87$ | $73.3 \pm 6.21$ | $59.8 \pm 4.72$ | $63.4 \pm 5.13$ |
| | MBSSFCC (Jiang et al., 2022) | $89.5 \pm 6.74$ | $78.7 \pm 6.75$ | $76.2 \pm 3.64$ | $74.3 \pm 7.04$ |
| | DBPNet (Ni et al., 2024) | $95.3 \pm 3.50$ | $80.2 \pm 6.79$ | $62.2 \pm 6.27$ | $63.3 \pm 4.56$ |
| | DARNet (Yan et al., 2024) | $95.5 \pm 4.89$ | $81.2 \pm 6.34$ | $91.3 \pm 2.73$ | $87.6 \pm 13.19$ |
| | SSF-DST$^*$ (Zhu et al., 2025) | $96.6 \pm 4.11$ | $85.0 \pm 6.00$ | $57.0 \pm 6.72$ | $60.2 \pm 6.84$ |
| | MHANet (Li et al., 2025) | $96.6 \pm 3.67$ | $83.0 \pm 7.14$ | $92.9 \pm 3.93$ | $92.0 \pm 3.84$ |
| | **PCRNet (ours)** | $\mathbf{98.8 \pm 1.70}$ | $\mathbf{85.6 \pm 7.38}$ | $\mathbf{94.3 \pm 3.24}$ | $\mathbf{94.2 \pm 3.10}$ |

consecutive epochs, training is stopped. Additionally, we adopt the Adam optimizer with a learning rate of 4e-3 and weight decay of 3e-4 to train the model.

Before feeding the EEG data into PCRNet, we employ the CSP algorithm to extract raw features from the EEG data under different brain states, then rearrange them into $\tilde{E} \in \mathbb{R}^{64 \times 128}$ as the network input. The network first utilizes the TCC module with 16 attention heads to calibrate temporal dependencies. Subsequently, the DDI module processes the features with a base dimension of 16. Specifically, the temporal stream in the DDI module employs dilated convolutions with a fixed kernel size of $(1, 3)$ and exponential dilation rates of $d \in \{1, 2, 4\}$. The refined features are then passed through specific convolutional layers including a temporal convolution layer (kernel size $(1, 2)$) and a spatial convolution layer (kernel size $(64, 1)$), followed by global average pooling. These operations further aggregate temporal and spatial cues for the final decision while improving representation compactness and robustness. Finally, a fully connected layer (input: 5, output: 2) produces the binary

AAD output $p$.

## 4. Results

### 4.1. Comparative Analysis

In this work, we compare the performance of our PCRNet with other advanced models, as detailed in Table 2. The results demonstrate that PCRNet achieves a significant advancement over the current SOTA methods.

On the KUL dataset, PCRNet achieves average accuracies of 98.4% (SD: 2.83), 98.5% (SD: 2.35), 98.8% (SD: 1.70%) under 0.1-second, 1-second and 2-second decision windows, respectively. Compared with the current SOTA model, PCRNet achieves an improvements of 2.8%, 2.7%, and 2.2%, respectively.

On the DTU dataset, PCRNet achieves average accuracies of 78.6% (SD: 7.77%), 85.3% (SD: 7.27%), 85.6% (SD: 7.38%) under 0.1-second, 1-second and 2-second decision windows, respectively. Compared with the current SOTA

*Table 3.* Ablation Study on KUL, DTU, and AVED dataset. The KUL and DTU datasets contain the audio-only scene. The AVED dataset contains both audio-only (AO) and audio-visual (AV) scenes. TCC represents the temporal context calibration module. MTA denotes the multi-scale temporal attention block. DDI represents the dual-domain integration module. RSI denotes the residual-spectral interface block.

| Decision Window | Model | Dataset | | | |
|---|---|---|---|---|---|
| | | KUL | DTU | AVED (AO) | AVED (AV) |
| 0.1-second | w/o TCC | 93.5 ± 5.26 | 77.4 ± 6.92 | 61.8 ± 1.85 | 60.3 ± 1.90 |
| | w/o MTA | 97.7 ± 3.29 | 78.2 ± 7.81 | 69.0 ± 1.39 | 68.6 ± 2.34 |
| | w/o DDI | 95.6 ± 4.09 | 76.7 ± 8.08 | 70.0 ± 1.58 | 69.1 ± 3.02 |
| | w/o RSI | 98.3 ± 2.74 | 77.5 ± 8.22 | 69.2 ± 1.63 | 68.6 ± 2.53 |
| | **PCRNet (ours)** | **98.4 ± 2.83** | **78.6 ± 7.77** | **70.2 ± 1.72** | **69.2 ± 2.65** |
| 1-second | w/o TCC | 94.8 ± 4.37 | 80.3 ± 6.96 | 75.2 ± 4.98 | 74.4 ± 3.60 |
| | w/o MTA | 97.8 ± 3.02 | 85.1 ± 7.17 | 85.7 ± 11.08 | 86.5 ± 10.09 |
| | w/o DDI | 97.2 ± 3.75 | 82.8 ± 7.91 | 85.7 ± 3.45 | 86.5 ± 4.05 |
| | w/o RSI | 98.0 ± 3.07 | 83.9 ± 8.88 | 86.6 ± 13.03 | 89.1 ± 3.75 |
| | **PCRNet (ours)** | **98.5 ± 2.35** | **85.3 ± 7.27** | **89.7 ± 3.49** | **89.5 ± 3.16** |
| 2-second | w/o TCC | 94.9 ± 4.97 | 80.0 ± 5.53 | 75.2 ± 7.87 | 78.9 ± 7.55 |
| | w/o MTA | 98.2 ± 2.59 | 85.0 ± 8.87 | 90.6 ± 7.64 | 94.1 ± 3.81 |
| | w/o DDI | 98.7 ± 1.44 | 79.7 ± 10.00 | 86.3 ± 5.20 | 84.4 ± 8.37 |
| | w/o RSI | 98.2 ± 2.83 | 83.7 ± 8.99 | 90.6 ± 7.77 | 89.4 ± 9.87 |
| | **PCRNet (ours)** | **98.8 ± 1.70** | **85.6 ± 7.38** | **94.3 ± 3.24** | **94.2 ± 3.10** |

model, PCRNet achieves an improvements of 3.1%, 3.1%, and 2.6%, respectively.

On the AVED dataset, PCRNet achieves average accuracies of 70.2% (SD:1.72%), 89.7% (SD: 3.49%), 94.3% (SD: 3.24%) under 0.1-second, 1-second and 2-second decision windows, respectively in the audio-only scenario, and 69.2% (SD: 2.65%), 89.5% (SD: 3.16%), 94.2% (SD: 3.10%) under 0.1-second, 1-second and 2-second decision windows, respectively in the audio-visual scenario.

Overall, the outstanding performance of PCRNet across different datasets and decision windows suggests the potential of our model as an efficient method for its applications in real-world scenarios. Specifically on the KUL dataset, the model exhibits exceptional improvement under challenging ultra-short decision windows, validating its suitability for low-latency applications like neuro-steered hearing aids.

## 4.2. Ablation Analysis

We conduct extensive ablation experiments by removing the TCC, MTA, DDI and RSI modules to ensure a thorough analysis of our model. All experimental conditions are kept the same as in the previous settings. The results of these ablation experiments are shown in Table 3. These results validate the individual contribution of each component and confirms that their synergistic integration is key to PCRNet's superior performance.

The removal of the TCC along with a parallel MTA weakens the ability to synthesize robust spatiotemporal representations. The effectiveness of TCC can be attributed to its ability to aggregate temporal dependencies over multiple

receptive fields, allowing the network to identify both long-range and short-range temporal patterns across different dimensions within EEG signals while effectively suppressing irrelevant electrodes to focus on informative brain regions. Consequently, the removal of the TCC significantly impairs the model's decoding performance.

The removal of the DDI leads to the loss of the model's ability to bridge the time and frequency domains effectively, reducing its discriminative capability and making it more susceptible to the inherent noise contamination of EEG signals. This result confirms the effectiveness of DDI in capturing long-range temporal contexts and suppress noise-dominated frequencies through parallel temporal and spectral branches. Finally, removing RSI leads to a moderate performance decrease in several settings, suggesting that RSI acts as a complementary refinement component within the dual-domain framework. Although the remaining temporal and dual-domain components still retain substantial discriminative capacity, RSI further improves phase-sensitive spectral refinement and helps enhance feature discriminability in the complex domain. This result indicates that the benefits of PCRNet arise from the coordinated interaction between components.

## 4.3. Visualization Analysis

To investigate the interpretability of the learned spatial attention mechanism, we visualize the spatial distribution of the attention weights across the scalp. Since the model is trained to identify the attended speech source, the spatial weights essentially reflect the importance of different brain regions in the decision-making process.

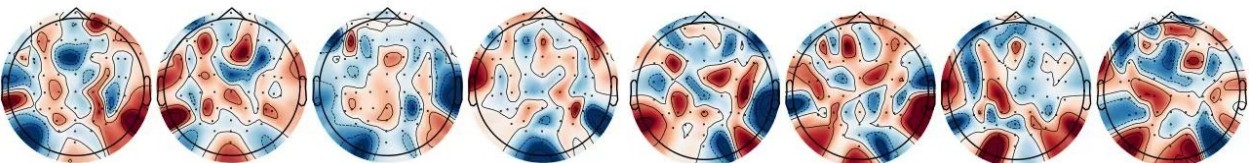

*(a)* Topography maps of the decoder weights associated with the EEG electrodes on KUL dataset for two individual subjects

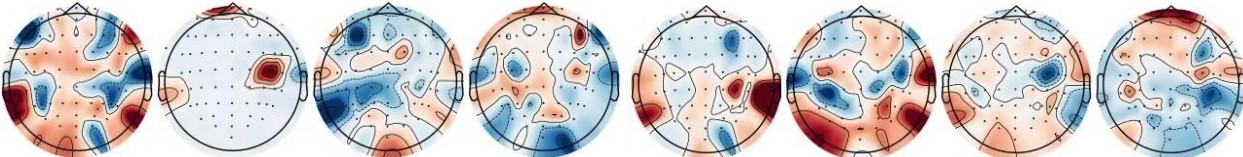

*(b)* Topography maps of the decoder weights associated with the EEG electrodes on DTU dataset for two individual subjects

*Figure 2.* Visualization of spatial attention weights. The maps reveal that the model automatically focuses on neurophysiologically relevant areas, with higher weights (red) predominantly concentrated over the temporal and parietal lobes, which are known to be associated with auditory processing and selective attention.

Specifically, we aggregate and average the masking weights generated by the spatial attention module over all 1-second decision windows across some subjects. The resulting topography maps for the KUL and DTU datasets are illustrated in Figure 2. In these maps, the black dots represent all 64 EEG electrodes, and the color intensity indicates the magnitude of the learned weights, with the red color corresponding to a higher weight.

As shown in Figure 2a, the KUL dataset exhibits relatively diverse spatial patterns across subjects. Several maps show evident activity variations over lateral temporal and parietal regions, which are broadly consistent with the involvement of auditory and attentional processing in speech listening tasks. At the same time, noticeable inter-subject variability can be observed, with some subjects showing stronger posterior or frontal distributions. This variability is reasonable for EEG-based auditory attention datasets, since scalp-level responses are affected by individual differences, electrode impedance, head geometry, and trial-level signal fluctuations. Therefore, the KUL topographies suggest that although auditory-attention-related spatial patterns are present, the dataset also contains substantial subject-dependent spatial heterogeneity.

As shown in Figure 2b, the DTU dataset presents comparatively more distributed and spatially variable scalp patterns. Several subjects exhibit clear lateralized responses over temporal or temporoparietal areas, while others show broader frontal, central, or posterior distributions. Compared with KUL, the DTU maps appear less uniform across subjects, which may reflect differences in recording conditions, preprocessing, task design, or subject-specific EEG characteristics. The presence of bilateral temporal and parietal variations is still compatible with auditory speech processing and attentional modulation, but the overall pattern should be

interpreted as descriptive evidence of dataset-level spatial characteristics rather than precise localization of neural sources.

Overall, the scalp maps of KUL and DTU show that both datasets contain spatially structured EEG responses related to auditory attention tasks, while also exhibiting clear inter-subject variability. Such spatial variability also supports the motivation of recent graph- and hypergraph-based AAD studies that explicitly model inter-channel spatial dependencies (Fan et al., 2024a; Zhou et al., 2025). These visualizations provide a qualitative overview of the spatial distribution of EEG activity in the datasets. Since the maps are computed directly from EEG signals rather than from model outputs, they should not be interpreted as evidence of learned attention mechanisms or module-specific behavior. Instead, they serve as supplementary visualization of the dataset characteristics and help illustrate the spatial variability present in EEG-based auditory attention decoding.

## 5. Conclusion

This paper proposes a novel phase-aware complex refinement network named PCRNet to address the critical oversight of phase information and the susceptibility to spectral noise in existing AAD methods. First, the TCC utilizes a parallel MTA-guided gating mechanism to calibrate multiscale temporal dependencies. Then, the DDI further integrates a cascade of dilated convolutions to model long-range temporal contexts with and an RSI block for fine-grained spectral processing. By dynamically refining real and imaginary components separately, RSI effectively recovers phase information and suppresses noise-dominated frequencies, thereby distinguishing structured neural patterns from background artifacts. Subsequently, specific convolutional layers

are employed to aggregate expressive EEG features and enhance the robustness of the model. We evaluate the performance of the proposed PCRNet on three datasets: KUL, DTU, and AVED. The model exhibits exceptional stability and accuracy, particularly under challenging ultra-short decision windows, validating its suitability for low-latency applications such as neuro-steered hearing aids. In future research, we plan to further explore PCRNet's performance on cross-subject and cross-domain tasks to enhance its generalization ability across different acoustic environments and recording setups.

## Impact Statement

Our work aims to advance EEG-based AAD by proposing PCRNet, a Phase-aware Complex Refinement Network that improves decoding robustness and efficiency over existing methods. By maintaining strong performance, particularly under challenging ultra-short decision windows, PCRNet may promote research on low-latency applications such as neuro-steered hearing aids and other assistive systems that selectively enhance an attended speaker in complex acoustic environments, potentially reducing listening effort and improving communication for target users.

However, there are important ethical and societal considerations associated with any method that decodes cognitive attentional states from EEG signals. EEG recordings are highly sensitive, and AAD systems could, in principle, reveal aspects of users' preferences or mental states beyond what they intend to disclose, especially if deployed without strong consent and privacy protections or misused for surveillance and monitoring. Our empirical evaluation is conducted on a limited number of laboratory datasets under controlled conditions, so generalization to broader populations, languages, recording setups, and clinical groups remains uncertain and may lead to unequal performance. Moreover, training and large-scale evaluation of neural AAD models consume computing resources and energy.

Overall, we expect our contributions to have a positive impact when used in carefully designed, user-centered assistive or research applications that obtain informed consent, protect the confidentiality of EEG data, and operate under appropriate regulatory and ethical oversight. We encourage future work to broaden dataset diversity, monitor bias and failure modes, and develop transparent, interpretable, and user-controllable AAD systems so that the potential benefits of AAD can be realized while the associated risks are responsibly managed.

## Acknowledgements

This work is supported by the Brain Science and Brain-like Intelligence Technology-National Science and Technology Major Project (No. 2021ZD0201500), the National Natural Science Foundation of China (NSFC) (No.62571002, 62476004, 62236002), Excellent Youth Foundation of Anhui Scientific Committee (No. 2408085Y034), Cloud Ginger XR-1.

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

## A. Motivation Illustration

To intuitively demonstrate the motivation and working principle of the DDI module, Figure 3 presents a visual decomposition of the processing of a representative EEG signal.

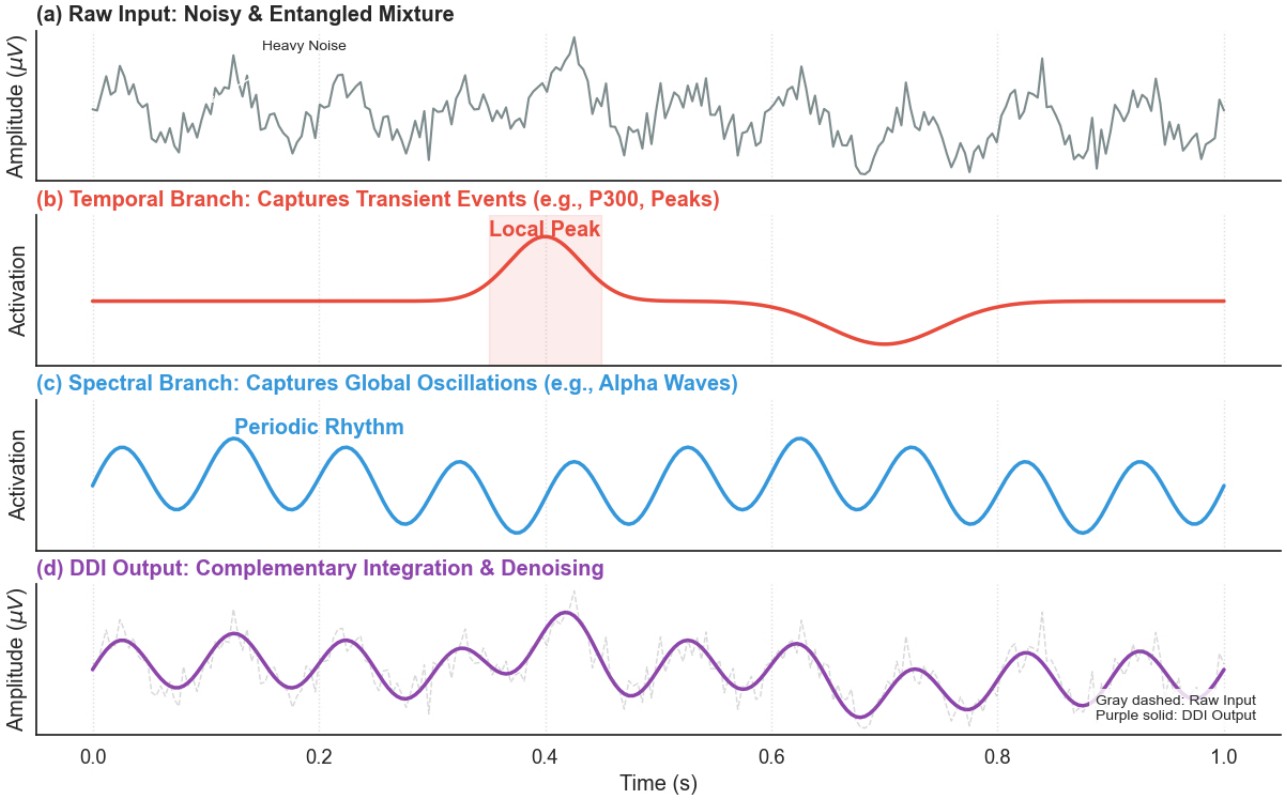

*Figure 3.* Motivation Illustration for Dual-Domain Complementary Feature Learning

As shown in Figure 3(a), raw EEG signals are typically highly entangled mixtures containing transient evoked potentials (e.g., P300), background rhythms (e.g., Alpha waves), and heavy random noise. Traditional single-domain convolutions often struggle to capture these distinct characteristics simultaneously.

As illustrated in Figure 3(b), the temporal branch focuses on capturing transient events and local peaks, effectively acting as a filter for periodic background fluctuations. As illustrated in Figure 3(c), the spectral branch focuses on extracting global oscillations and stationary rhythms. Crucially, the proposed DDI explicitly incorporates phase information. While magnitude reflects signal energy, phase information encodes the critical temporal alignment and structural semantics of the EEG waveform. By preserving phase dynamics, this branch ensures that the synthesized features maintain precise synchronization with the original physiological events, showing robustness against local abrupt changes.

Overall, the result in 3(d) demonstrates the effectiveness of fusing these complementary views. By integrating the transient-sensitive temporal features with phase-aware spectral representations, the DDI module generates a denoised, high-quality feature representation. This dual-domain phase-guided strategy not only enhances the semantic richness of the features but also ensures structural fidelity, significantly improving the model's robustness in low signal-to-noise ratio environments.

.

## B. Detailed Algorithm of PCRNet

The detailed inference process of our proposed PCRNet is outlined in Algorithm 1. The framework takes raw EEG signals $\mathbf{X}$ as input and processes them through three key stages. $\odot$ denotes Element-wise multiplication. $\Re(\cdot)$ and $\Im(\cdot)$ represents Real and Imaginary parts of a complex number. $Conv1d_k$ refers to a 1D Convolution with kernel size $k$.

---

**Algorithm 1** Forward Propagation of PCRNet

---

**Require:** EEG Signals $\mathbf{X} \in \mathbb{R}^{B \times 1 \times C \times T}$
**Ensure:** Predicted Class Probabilities $\mathbf{Y}$
 1: **Hyperparameters:** Window length $T$, EEG channels $C$, Base dim $d$.
 2: {**Phase 1: Temporal Context Calibration (TCC)**}
 3: Permute $\mathbf{X} \rightarrow \mathbf{X}_{perm} \in \mathbb{R}^{B \times C \times 1 \times T}$
 4: Compute Projections: $\mathbf{Q}, \mathbf{K}, \mathbf{V} \leftarrow \text{Conv2d}(\mathbf{X}_{perm})$
 5: **Function** MTA($\mathbf{Z}$): {Multi-scale Temporal Attention}
 6:     $\mathbf{Z}' \leftarrow \text{SpatioConv}(\mathbf{Z})$
 7:     Split $\mathbf{Z}'$ into chunks $\mathbf{z}_1, \mathbf{z}_2, \mathbf{z}_3$
 8:     Apply kernels $k \in \{2, 4, 6\}$: $attn \leftarrow \sum \text{Conv1d}_k(\mathbf{z}_i)$
 9:     **Return** Sigmoid($attn$)
10: **End Function**
11: Modulate Projections:
12:     $\mathbf{Q} \leftarrow \mathbf{Q} \odot (1 + \text{MTA}(\mathbf{Q}))$
13:     $\mathbf{K} \leftarrow \mathbf{K} \odot (1 + \text{MTA}(\mathbf{K}))$
14:     $\mathbf{V} \leftarrow \mathbf{V} \odot (1 + \text{MTA}(\mathbf{V}))$
15: Compute Self-Attention with Learnable Temperature $\tau$:
16:     $\mathbf{A} \leftarrow \text{Softmax}\left(\tau \cdot \frac{\mathbf{Q}\mathbf{K}^T}{\sqrt{d_{head}}}\right)$
17:     $\mathbf{X}_{tcc} \leftarrow \text{Proj}_{out}(\mathbf{A}\mathbf{V})$
18: Permute $\mathbf{X}_{tcc} \leftarrow \mathbf{X}_{tcc}.\text{permute}(...)$ {Back to $B \times 1 \times C \times T$}
19: {**Phase 2: Dual Domain Integration (DDI)**}
20: $\mathbf{X}_{emb} \leftarrow \text{Preprocess}(\mathbf{X}_{tcc})$
21: *Time Stream:*
22:     $\mathbf{H}_{time} \leftarrow \mathbf{X}_{emb}$
23:     **for** $r \in \{1, 2, 4\}$ **do**
24:         $\mathbf{H}_{time} \leftarrow \text{DilatedConv}(\mathbf{H}_{time}, \text{rate} = r) + \mathbf{H}_{time}$
25:     **end for**
26: *Freq Stream (Gated Spectral Refinement):*
27:     $\mathbf{F} \leftarrow \text{FFT2d}(\text{LayerNorm}(\mathbf{X}_{emb}))$
28:     Magnitude Mask $\mathbf{M} \leftarrow \text{Sigmoid}(\text{MLP}(|\mathbf{F}|))$
29:     Refine Real/Imag: $\mathbf{F}'_{real} \leftarrow \text{Mixer}(\Re(\mathbf{F})) \odot \mathbf{M}$
30:     Refine Real/Imag: $\mathbf{F}'_{imag} \leftarrow \text{Mixer}(\Im(\mathbf{F})) \odot \mathbf{M}$
31:     $\mathbf{H}_{freq} \leftarrow \text{iFFT2d}(\mathbf{F}'_{real} + i\mathbf{F}'_{imag})$
32: Fusion: $\mathbf{X}_{ddi} \leftarrow \mathbf{X}_{tcc} + \text{Proj}(\text{Concat}(\mathbf{H}_{time}, \mathbf{H}_{freq}))$
33: {**Phase 3: Feature Extraction & Output**}
34: $\mathbf{X}_{feat} \leftarrow \text{Conv2d}_{temporal}(\mathbf{X}_{ddi})$
35: $\mathbf{X}_{feat} \leftarrow \text{Conv2d}_{spatial}(\mathbf{X}_{feat})$
36: $\mathbf{Y} \leftarrow \text{Linear}(\text{Flatten}(\text{AvgPool}(\mathbf{X}_{feat})))$
37: **return** $\mathbf{Y}$

---

## C. Attention Mechanism Analysis

To further investigate the internal decision-making process of PCRNet, we visualize the attention weight across different decision windows. As illustrated in Figure 4, it reveals a distinct diagonal-dominant structure across all three temporal scales. The prominent dark blue diagonal bands indicate that the model maintains a robust focus on local temporal contexts across scales, effectively capturing the continuous evolutionary characteristics of EEG signals while suppressing irrelevant long-range noise. The off-diagonal regions are predominantly pale yellow (near-zero weights), demonstrating the model's ability to suppress irrelevant long-range dependencies. By filtering out distant and unrelated time points, the attention mechanism effectively reduces temporal noise and focuses computational resources on the valid signal envelope.

In the 0.1-second window (Figure 4a), the discrete, block-like diagonal highlights the model's high-fidelity focus on

immediate transient features at a granular level. In the 1-second and 2-second windows (Figure 4b,c), as the sequence length increases, the attention band remains smooth and continuous. This confirms that the PCRNet adaptively scales its receptive field, maintaining a consistent window of temporal focus regardless of the input duration.

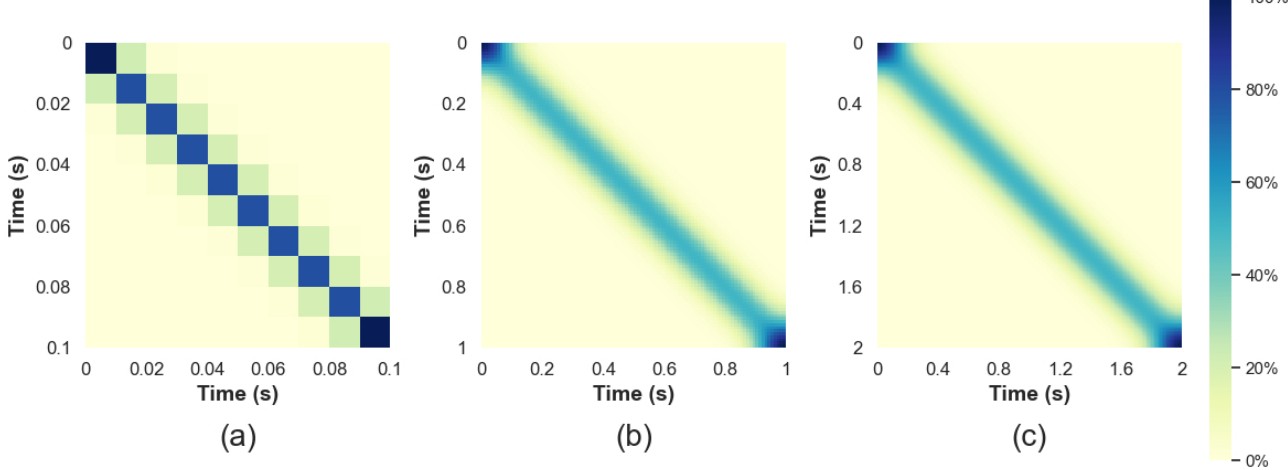

*Figure 4.* Visualization of average temporal attention weights of PCRNet across different decision windows. Each row represents the attention weights of a specific EEG signal in the sequence with all signals. (a) The average attention weights for 0.1-second. (b) The average attention weight for 1-second. (c) The average attention weight for 2-second.

## D. Computational Complexity Analysis

As summarized in Table 4, we compared the trainable parameters of PCRNet with several SOTA AAD models. The results indicate that PCRNet contains only approximately 0.03 M (measured at 30.86 k) trainable parameters, achieving a substantial reduction in model size compared to traditional architectures. Specifically, the parameter count of PCRNet is roughly 140 times smaller than that of SSF-CNN (4.21 M) and nearly 2,800 times smaller than the high-complexity MBSSFCC (83.91 M). Even when compared to recent lightweight models such as DBPNet (0.91 M) and DARNet (0.08 M), PCRNet reduces the parameter footprint by approximately 30 times and 2.6 times, respectively. While MHANet (0.02 M) represents an extremely minimalist design, PCRNet introduces a more robust dual-domain integration module with a marginal increase of only 0.01 M parameters, enabling more sophisticated spatiotemporal and spectral feature interactions at a negligible cost.

*Table 4.* Comparison of Trainable Parameters across different AAD models.

| Model | Trainable Parameters |
|---|---|
| SSF-CNN (Cai et al., 2021) | 4.21 M |
| MBSSFCC (Jiang et al., 2022) | 83.91 M |
| DBPNet (Ni et al., 2024) | 0.91 M |
| DARNet (Yan et al., 2024) | 0.08 M |
| MHANet (Li et al., 2025) | 0.02 M |
| **PCRNet (ours)** | **0.03 M** |

To evaluate the runtime efficiency for AAD tasks, we measured the computational load for a standard 1-second EEG window. The total FLOPs is 95.05 MFLOPs and the total model architecture complexity (MAC) is 47.52 MMACs. The computational load of 95.05 MFLOPs is remarkably low for a model that integrates both time and frequency domain features. The inclusion of RSI, which relies on the FFT, ensures that global spectral features are captured with $O(NlogN)$ complexity, avoiding the $O(N^2)$ quadratic cost associated with standard Transformer-based self-attention.

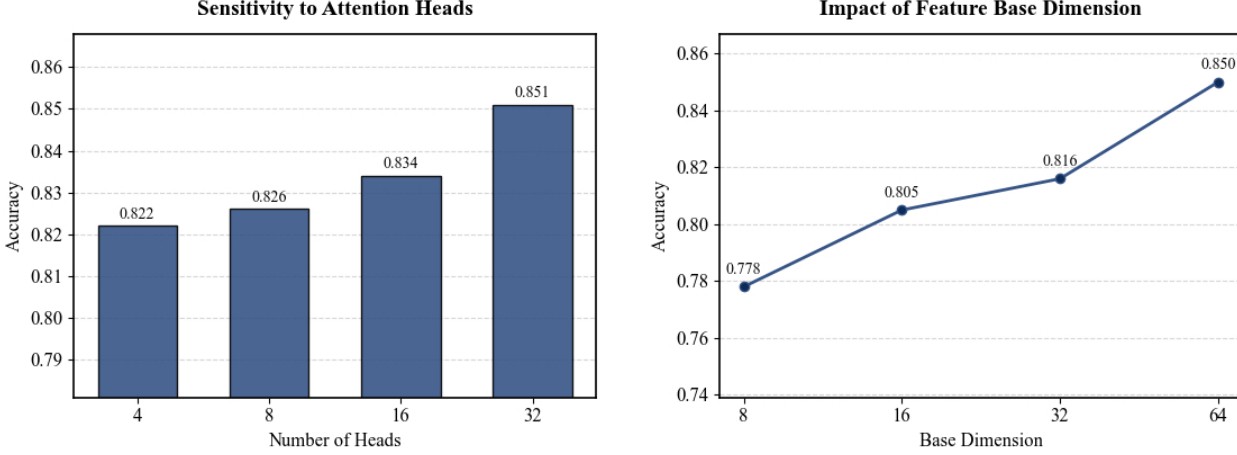

*Figure 5.* Impact of Hyperparameters on AAD Accuracy. (Left) Sensitivity analysis of the number of attention heads ($N_h$). (Right) Influence of the feature base dimension ($d_{base}$) on decoding performance. The results demonstrate a consistent performance gain as the model complexity increases within the evaluated range.

## E. Hyperparameters Analysis

To ensure the optimal performance and robustness of PCRNet in the AAD task, we conducted a comprehensive sensitivity analysis on two core hyperparameters: the number of attention heads in the TCC module and the base dimension of the DDI module. This analysis aims to investigate how model complexity and feature representation capacity influence the decoding of neural tracking of speech. All experiments were performed using the same cross-validation protocol to ensure fair comparison. The experimental results in Figure 5 illustrate the correlation between model architecture complexity and AAD performance.

As shown in the left bar chart, the decoding accuracy exhibits a steady upward trend as the number of attention heads ($N_h$) increases from 4 to 32. This improvement suggests that a multi-head mechanism with higher granularity allows the TCC module to model more diverse and sophisticated cross-channel relational patterns. In the context of AAD, this enhanced representational power facilitates a more precise extraction of neural tracking features associated with the attended speaker.

The right line plot depicts the influence of the feature base dimension ($d_{base}$). We observe a significant performance gain, with accuracy rising from approximately 0.78 to 0.85 as expands from 8 to 64. This indicates that increasing the internal channel capacity of the dual-domain streams substantially boosts the model's ability to represent non-linear neural responses to speech. While a larger dimension leads to superior accuracy, it also introduces higher computational demands.

Therefore, considering both accuracy and computational demands, $N_h = 16$ and $d_{base} = 32 \ (or \ 16)$ are identified as favorable configurations that strike a balance between efficiency and high-accuracy decoding.

