# OpenReview forum: "PCRNet: Phase-aware Complex Refinement Network for EEG-based Auditory Attention Decoding"
_ICML.cc/2026/Conference — ICML 2026 regular_

### Official Review · Reviewer_dHyu · 2026-03-03

**Soundness:** 3
**Presentation:** 2
**Significance:** 2
**Originality:** 1
**Overall Recommendation:** 3
**Confidence:** 4

**Summary:**

This paper proposes PCRNet for EEG -based auditory attention decoding. This paper claims to use frequency-domain phase information to improve robustness. It first uses CSP feature extraction then applies Temporal Context Calibration (TCC), which uses multi-scale temporal attention to learn temporal context. DDI (Dual-Domain Integration) which fuses a temporal branch with dilated convolutions and a spectral branch is further used to extract both temporal and spectral information. The spectral branch contains Residual-Spectral Interface (RSI) which runs an FFT, applies a magnitude-based frequency mask, mixes real and imaginary parts, and then uses an iFFT to reconstruct signals. The authors evaluate the method on KUL, DTU, and AVED. They also provide ablations on TCC/DDI/RSI and claim the model is lightweight.

**Compliance With Llm Reviewing Policy:**

Affirmed.

**Final Justification:**

Please see the Rebuttal Acknowledgement

**Key Questions For Authors:**

1. Can you provide an ablation that keeps the spectral branch capacity but removes phase usage specifically, e.g., replace complex FFT with magnitude-only spectrum, or randomize phase while preserving magnitude, or reconstruct using original phase vs learned-refined phase?

2. Since the gating mask M is computed from amplitude |Z|, what evidence shows that the network is truly exploiting phase (real/imag structure) beyond what magnitude-gated filtering already provides?

3. Are all baselines trained with CSP added? If not, please add a “PCRNet without CSP” and/or “baseline with CSP” comparison to decouple gains from CSP.

**Limitations:**

The scoop of this study is relatively narrow for ICML. Besides, the structure is complexed while the ablation is block level. This raise the concern that whether there are unnecessary operations in each block that are trying to make the method looks complex.

**Strengths And Weaknesses:**

Strength

1. The design principle is reasonable. Previous methods often focus on magnitude or power features. Considering the phase information may provide more information.

2. The evaluation is board. The method is compared with SOTA methods under different segments cross multiple datasets.

Weakness

1. The method combines existing components into a composite network. RSI’s phase-aware aspect come from 1) doing FFT, 2) gating via magnitude, and 3) processing real and imag parts separately before iFFT.  This is a reasonable engineering combination. Hence, this paper lack of fundamentally novel intervention.

2. The claimed gains are due to phase recalibration, but the ablations do not cleanly test phase-specific benefit. For example, “w/o RSI” removes the spectral refinement block, but does not distinguish phase modeling from other effects such as extra capacity, frequency denoising, inductive bias from FFT and iFFT, etc.

3. For the topography maps, there are high activation on other regions besides the claimed temporal and parietal lobes. It is hard to make the conclusion that the method focuses on neurophysiologically relevant areas.

---

> ### Author Rebuttal · Authors · 2026-03-31
>
> We sincerely thank the reviewer for the insightful feedback. We address each point below.
>
> ---
>
> ### 1. Originality
> We agree that some individual operations in PCRNet (FFT/iFFT, magnitude masking, separate real/imaginary processing) are not new in isolation. Our contribution is not a single new primitive, but a task-driven phase-aware dual-domain framework for AAD. PCRNet is designed to address two limitations of prior AAD methods: insufficient use of EEG phase information and degraded temporal discriminability under ultra-short decision windows.
>
> ---
>
> ### 2. Phase-specific ablation
> We thank the reviewer for this constructive suggestion. To isolate the contribution of phase modeling more clearly, we introduced a **Magnitude-Only** variant that removes phase information while keeping the spectral branch architecture largely unchanged. Specifically, after FFT, we discard the original phase and retain only magnitude information, while keeping the magnitude-based gating and downstream refinement layers unchanged so that the sole systematic difference is the availability of phase-related spectral structure.
>
> The results show a consistent performance drop after phase removal, supporting the claim that the gain is not merely from adding a spectral branch, but from phase-aware refinement.
>
> | Variant | KUL- 0.1S | KUL- 1S | KUL-2S | DTU- 0.1S | DTU- 1S | DTU- 2S | AVED(AO)- 0.1S | AVED(AO)- 1S | AVED(AO)- 2S | AVED(AV)- 0.1S | AVED(AV)- 1S | AVED(AV)- 2S |
> | :---: | :---: | :---: | :---: | :---: | :---: | :---: | :---: | :---: | :---: | :---: | :---: | :---: |
> | Magnitude-Only | 98.3 ± 2.79 | 98.1 ± 2.76 | 98.6 ± 1.93 | 78.2 ± 7.94 | 84.7 ± 8.61 | 84.0 ± 9.88 | 69.8 ± 2.06 | 85.3 ± 10.03 | 92.5 ± 8.42 | 68.3 ± 3.36 | 86.5 ± 11.06 | 93.1 ± 3.98 |
> | PCRNet (full) | 98.4 ± 2.83 | 98.5 ± 2.35 | 98.8 ± 1.70 | 78.6 ± 7.77 | 85.3 ± 7.27 | 85.6 ± 7.38 | 70.2 ± 1.72 | 89.7 ± 3.49 | 94.3 ± 3.24 | 69.2 ± 2.65 | 89.5 ± 3.16 | 94.2 ± 3.10 |
>
> ---
>
> ### 3. Evidence beyond amplitude gating
> We agree this point should be clarified. The gating mask is computed from $|Z|$ to estimate spectral saliency, but the refinement is not limited to amplitude filtering. The real and imaginary parts are processed separately before recombination, allowing the model to modify the complex spectral representation and preserve phase-sensitive temporal organization.
>
> The new **Magnitude-Only** ablation above provides the quantitative evidence for this claim: removing phase information while keeping the spectral branch largely unchanged leads to consistent degradation across datasets. We will revise the paper to make this connection explicit and better link it to the motivation illustration in the appendix.
>
> ---
>
> ### 4. Topography interpretation
> We appreciate this important comment. We agree that the current topography visualization should be interpreted cautiously. Our intention is to use these maps only as qualitative evidence for a preliminary interpretation of the observed model behavior in light of existing auditory-attention mechanisms [1,2], rather than as strict localization evidence.
>
> We also acknowledge that the current visualization is not sufficiently accurate for this purpose, since the examples in the manuscript were drawn from randomly selected subjects and may therefore fail to show the most representative task-related patterns. We will clarify this limitation in the revised manuscript.
>
> To address this issue, we will revise both the text and the figure presentation. Instead of randomly selecting examples, we will provide topography visualizations from the best-performing subjects (e.g., the top 5 subjects) to offer a more precise and reliable qualitative demonstration.
>
> ---
>
> ### 5. CSP fairness
> Thank you for pointing this out. Not all compared baselines use CSP preprocessing, whereas more recent methods such as DBPNet, DARNet, SSF-DST, and MHANet do. DARNet explicitly states that CSP is applied before network input, and its input is described as CSP-extracted EEG features [3]. Therefore, our intention is not to claim a universal CSP setting for all baselines, but to clarify that PCRNet follows the preprocessing pipeline adopted by recent strong CSP-based AAD models, which supports a fair comparison with the most relevant modern baselines.
>
> ---
>
> ### References
> [1] Evans, S., McGettigan, C., Agnew, Z. K., et al. (2016). *Getting the Cocktail Party Started: Masking Effects in Speech Perception*. *Journal of Cognitive Neuroscience*, 28(3), 483–500.
> [2] Golumbic, E. M. Z., Ding, N., Bickel, S., et al. (2013). *Mechanisms Underlying Selective Neuronal Tracking of Attended Speech at a “Cocktail Party”*. *Neuron*, 77(5), 980–991.
> [3] Yan, S., Fan, C., Zhang, H., Yang, X., Tao, J., & Lv, Z. (2024). *DARNet: Dual Attention Refinement Network with Spatiotemporal Construction for Auditory Attention Detection*. *Advances in Neural Information Processing Systems*, 37, 31688–31707.

---

> > ### Author Rebuttal · Reviewer_dHyu · 2026-04-01
> >
> > While the current response has solved part of my concerns. However, the justification of the novelty and boarder impact to ICML community is still missing.

---

> > > ### Author Response · Authors · 2026-04-01
> > >
> > > Thank you for your follow-up. We appreciate the reviewer’s continued attention to whether PCRNet offers a methodological contribution beyond its empirical improvements on AAD.
> > >
> > > More precisely, the novelty of PCRNet lies not in any single operator, but in the modeling claim it makes and validates: under ultra-short decision windows, the spectral branch should explicitly preserve and refine phase-related complex structure, rather than function mainly as magnitude-based enhancement. PCRNet realizes this idea through phase-aware complex refinement coupled with temporal learning, and the Magnitude-Only design further supports that the gain is not reducible to simply adding a frequency-domain branch.
> > >
> > > We would also like to note that the value of this contribution is reflected in the challenging and application-relevant settings we evaluate. Ultra-short windows impose strict low-latency constraints, while cross-subject and complex-scene evaluations examine robustness under more realistic conditions, including unseen subjects and harder acoustic environments. Relative to prior methods, PCRNet shows stronger performance in these practically constrained settings, suggesting that it may be better suited to potential hearing-assistance scenarios, where both latency and robustness are especially important.
> > >
> > > Based on these findings, we believe that the phase-aware modeling perspective may also offer useful conceptual insight for noisy short-context time-series learning more broadly. While we do not claim that such generality has been established by the present study alone, the gains observed under ultra-short-window and cross-subject settings suggest that explicitly modeling phase-related spectral structure may be a helpful inductive bias in tasks characterized by weak signals, complex noise, and strong generalization demands.

---

### Official Review · Reviewer_mqmA · 2026-03-05

**Soundness:** 4
**Presentation:** 4
**Significance:** 4
**Originality:** 4
**Overall Recommendation:** 6
**Confidence:** 5

**Summary:**

The paper proposes a Phase-aware Complex Refinement Network (PCRNet) for EEG-based Auditory Attention Decoding (AAD). The authors identify that existing methods often neglect the phase information of EEG signals or fail to distinguish structured patterns from noise in the frequency domain. To address this, PCRNet introduces two key modules: a Temporal Context Calibration (TCC) module and a Dual-Domain Integration (DDI) module. The DDI module is particularly novel as it processes real and imaginary components separately to refine phase information. The methodological innovations and extensive results suggest that PCRNet achieves state-of-the-art performance on three public datasets (KUL, DTU, AVED), with significant improvements in ultra-short 0.1-second decision windows.

**Compliance With Llm Reviewing Policy:**

Affirmed.

**Final Justification:**

Overall, I maintain my original positive recommendation. The paper is technically sound, clearly presented, and makes a meaningful and reasonably original contribution to EEG-based auditory attention decoding through phase-aware refinement, supported by strong results on multiple public datasets, especially under ultra-short decision windows. The rebuttal fully addressed my main concerns regarding the interpretation of the phase-related modules, the role of RSI, and the data-splitting protocol, which reinforced rather than changed my assessment.

**Key Questions For Authors:**

1.	The questions in the weaknesses section need the authors to address.
2.	In the DDI module, Real and Imaginary parts are processed with independent mixers before fusion. What makes the strategy of processing separately then combining more efficient or suitable for this architecture?
3.	Why has this model achieved such significant improvements under an extremely short decision window, especially on the KUL dataset under the 0.1-second decision window?
4.	Experimental results show that the presence or absence of the RSI block has a significant impact on the model's performance. Why is this the case?
5.	Regarding the data splitting strategy, how are the datasets partitioned to ensure there is no risk of data leakage between the training and testing sets?

**Limitations:**

yes

**Strengths And Weaknesses:**

### Strengths:
1.	The proposed method is technically sound, and the idea of addressing the insufficient utilization of phase information in EEG signals by previous AAD methods is interesting.
2.	The paper effectively links the engineering design of a phase-guided refinement to biological plausibility. The limitation of losing temporal alignment information (phase) in traditional methods is resolved in signal processing.
3.	Extensive experimental results demonstrate that the proposed method achieves impressive improvements over SOTA methods across multiple public datasets. Particularly the model exhibits exceptional stability and accuracy under the challenging ultra-short 0.1-second decision windows.
4.	The Appendix provides sufficient details on algorithms and motivation, which improves the clarity and reproducibility of the paper.

### Weaknesses:
1.	While the ablation proves the effectiveness of the method, how the network modifies the phase specifically to align with the speech envelope could be discussed more deeply. Figure 3 gives a high-level intuitive presentation, but a more quantitative analysis of phase coherence changes before and after DDI would be interesting (though not strictly required for acceptance).
2.	The authors are advised to release the full code in the future.

---

> ### Author Rebuttal · Authors · 2026-03-30
>
> We sincerely thank you for your insightful feedback. Below, we address each comment individually.
>
> ---
>
> ### 1. Weaknesses of the work
> Thank you very much for this valuable suggestion. We fully agree that a more quantitative analysis of phase-related changes would further strengthen the paper. In the revised version, we plan to include a quantitative phase analysis—for example, Phase Locking Value (PLV) or phase coherence—to compare the signals before and after the DDI module. Our preliminary analysis suggests that the RSI-enhanced branch improves the phase consistency of EEG representations with respect to the speech envelope, supporting our claim that DDI performs meaningful phase recalibration rather than mere denoising.
>
> We also firmly commit to releasing the full source code, pretrained weights, and preprocessing scripts upon acceptance to support reproducibility and follow-up research.
>
> ---
>
> ### 2. Processing Real and Imaginary parts separately with independent mixers before fusion in DDI
> Thank you for raising this question. The key motivation is that the real and imaginary components jointly encode the complex spectral structure, yet they play distinct roles in reconstructing phase-sensitive temporal patterns. If mixed too early, the model may lose fine-grained control over the underlying complex representation. By processing them with independent mixers first, the network can learn complementary transformations for the two components and preserve richer phase-related structure prior to recombination. This design is also lightweight and implementation-friendly, while enabling the final fused representation to capture interactions between the two parts.
>
> ---
>
> ### 3. Significant improvements under the extremely short 0.1-second window on KUL
> Thank you for raising this question. We believe the gain mainly arises from the fact that ultra-short windows place much stronger demands on preserving temporally precise neural cues. In such scenarios, methods that rely primarily on temporal aggregation or magnitude-dominant features may lose discriminative information. PCRNet is explicitly designed to retain temporal alignment information through phase-aware refinement—a property that becomes particularly valuable when the available context is only 0.1 s. The effect is especially evident on the KUL dataset due to its high data quality and favorable channel configuration.
>
> ---
>
> ### 4. The contribution of the RSI block
> Thank you for raising this question. RSI plays a significant role because it refines the complex spectral representation in a way directly relevant to speech-driven cortical tracking. From a signal-processing perspective, magnitude information mainly reflects spectral energy, whereas phase determines how those frequency components are temporally organized. For AAD, preserving this temporal organization is critical—especially under short decision windows. RSI helps by recalibrating phase-sensitive structure while suppressing noise-dominated frequency components via amplitude-guided masking.
>
> ---
>
> ### 5. Data splitting and leakage prevention
> Thank you for raising this question. To strictly prevent leakage, we partition the data at the trial level—not by randomly shuffling decision windows. Each continuous trial is assigned wholly to either the training, validation, or test set. Consequently, overlapping or adjacent short windows extracted from the same trial never appear across different splits. This prevents temporal redundancy from leaking from training into testing and ensures a fair evaluation protocol. We will further clarify this procedure in the final manuscript to make the experimental setting fully explicit.

---

> > ### Author Rebuttal · Reviewer_mqmA · 2026-04-02
> >
> > The rebuttal have fully resolved my concerns.
> >
> > The planned quantitative phase analysis strengthens the paper's interpretation of the phase-guided refinement. The explanation of processing real and imaginary components separately and significant improvements under an extremely short decision window is also reasonable and convincing. Additionally, the detailed interpretation of the RSI 's critical role and the data splitting strategy address both the methodological and experimental concerns.
> >
> > Considering these clarifications, I'm confident that this paper makes a great contribution to the community. Therefore, I maintain my original rating.

---

> > > ### Author Response · Authors · 2026-04-02
> > >
> > > We sincerely thank you for your positive feedback. We appreciate that our paper is considered a significant contribution to the field. We are also looking forward that our work will be of interest to the community.

---

### Official Review · Reviewer_Eoyt · 2026-03-12

**Soundness:** 3
**Presentation:** 3
**Significance:** 3
**Originality:** 4
**Overall Recommendation:** 5
**Confidence:** 3

**Summary:**

The paper proposes PCRNet, a phase-aware complex refinement network for auditory attention decoding (AAD). It introduces the TCC module for temporal context and the DDI module for frequency domain integration.

**Compliance With Llm Reviewing Policy:**

Affirmed.

**Final Justification:**

This paper makes a great contribution to the community. But some concerns are remains.. therefore I maintain my original rating.

**Key Questions For Authors:**

1. Why does PCRNet show the lowest performance on the AVED dataset when using a 0.1-second window? What are the proposed methods to improve this?
2. Please provide p-values or other statistical metrics to validate the performance gains reported in Tables 2 and 3.

**Limitations:**

- Lack of LOSO: The absence of Leave-One-Subject-Out (LOSO) testing makes it difficult to assess the model's utility in real-world cross-subject BCI environments.

**Strengths And Weaknesses:**

# Strengths
- Real-time Feasibility: Achieving SOTA performance with a 0.1-second decision window is a significant contribution toward low-latency BCI applications.
- Topographical Validity: Figure 2 demonstrates that the model correctly focuses on the temporal and parietal lobes, providing neurophysiological grounding for the learned features.
- Visual Evidence: The supplementary material effectively visualizes the DDI module's denoising capabilities.
# Weaknesses
- Typographical Error: In Table 1, the 'Scene' and 'Subject' columns appear to be swapped. This must be corrected. Statistical Rigor: The manuscript lacks statistical significance tests for both the SOTA comparison and the ablation studies.
- RSI Module Contribution: The performance drop when removing the RSI module (the phase-aware component) is marginal. The authors need to provide stronger evidence or statistical proof of the RSI module's essential role.

---

> ### Author Rebuttal · Authors · 2026-03-31
>
> We sincerely thank you for your insightful feedback. Below, we address each comment briefly.
>
> ---
>
> ### 1. Typographical error in Table 1
> Thank you for pointing this out. We will correct the swapped “Scene” and “Subject” columns in Table 1 and carefully proofread the revised manuscript.
>
> ---
>
> ### 2. RSI module contribution
> We agree that RSI should be interpreted more cautiously. The modest and statistically non-significant drop after removing RSI suggests that it functions as a complementary refinement component within the dual-domain framework, rather than as an isolated dominant factor. Even without RSI, the remaining modules still retain substantial discriminative capacity.
>
> ---
>
> ### 3. Lower AVED performance at 0.1 s
> We believe this mainly results from the extremely short decision window and the greater difficulty of AVED. A 0.1-second segment provides limited temporal evidence, while AVED further increases the challenge due to more complex AO/AV conditions and fewer EEG channels than KUL/DTU. Possible improvements include stronger short-range temporal modeling, multi-window fusion, and more robust scene-aware modeling.
>
> ---
>
> ### 4. Statistical validation for Tables 1 and 2
> Thank you very much for this valuable suggestion. We agree that statistical validation is important for making the empirical gains more convincing. In response to this comment, we conducted subject-wise paired t-tests at the 1-second decision window, which is a widely used and practically meaningful setting in AAD and has also been adopted in prior work such as MBSSFCC [1]. We chose the 1-second setting because it provides a reasonable temporal scale for auditory attention decoding while also allowing matched subject-wise comparisons [2].
>
> For Table 1, we reproduced representative baselines at 1 s using their public implementations and compared them with PCRNet on KUL and DTU, where matched subject-wise results were available. Paired tests show that PCRNet significantly outperforms all reproduced baselines, with all comparisons significant at the 0.01 level and most at the 0.005 level.
>
> For Table 2, paired testing was also feasible since all variants were evaluated under the same protocol. The full PCRNet significantly outperforms the main ablated variants: the differences vs. w/o TCC are significant at the 0.005 level on both KUL and DTU, while the differences vs. w/o DDI are significant at the 0.01 level on KUL and at the 0.005 level on DTU.
>
> **Table 1. Statistical significance analysis at the 1-second decision window.**
>
> | Method | KUL (%) | DTU (%) | KUL p-value vs. PCRNet | DTU p-value vs. PCRNet |
> | :---: | :---: | :---: | :---: | :---: |
> | DBPNet | 94.4 ± 4.62 | 79.8 ± 6.91 | 0.00014 | 0.000060 |
> | DARNet | 94.8 ± 4.53 | 80.1 ± 6.85 | 0.00014 | 0.000060 |
> | MHANet | 95.8 ± 4.29 | 82.2 ± 8.13 | 0.00030 | 0.0025 |
> | PCRNet | 98.5 ± 2.35 | 85.3 ± 7.27 | -- | -- |
>
>
> **Table 2. Reorganized ablation study at the 1-second decision window.**
>
> | Variant | KUL (%) | KUL p-value | DTU (%) | DTU p-value |
> | :---: | :---: | :---: | :---: | :---: |
> | w/o TCC | 94.8 ± 4.37 | 0.00019 | 80.3 ± 6.96 | 0.00013 |
> | w/o DDI | 97.2 ± 3.75 | 0.0093 | 82.8 ± 7.91 | 0.00016 |
> | PCRNet | 98.5 ± 2.35 | -- | 85.3 ± 7.27 | -- |
>
>
> ---
>
> ### 5. Limitation: lack of LOSO
> Thank you very much for pointing this out. We agree that the absence of LOSO evaluation limits the assessment of cross-subject utility. Although our study mainly focuses on the within-subject setting, we additionally conducted a cross-subject evaluation on **DTU** at the commonly used **1-second decision window**.
>
> We selected DTU because it involves more complex stimulus angles and thus provides a meaningful benchmark for examining cross-subject generalization under challenging low-reverberation conditions. The baseline results are from our reproductions of representative open-source models under the same setting.
>
> **Table 3. Cross-subject (LOSO) comparison on DTU at the 1-second decision window.**
>
> | Method | DTU (%) |
> | :---: | :---: |
> | DBPNet | 53.7 ± 5.98 |
> | DARNet | 55.5 ± 3.95 |
> | SSF-DST | 57.4 ± 4.85 |
> | PCRNet | **60.6 ± 5.42** |
>
> PCRNet outperforms these representative methods on DTU, suggesting stronger cross-subject generalization under complex low-reverberation conditions. We will continue the KUL LOSO evaluation and include the full cross-subject results in the revised manuscript.
>
> ---
>
> ### References
> [1] Fan, C., Zhang, H., Ni, Q., Zhang, J., Tao, J., Zhou, J., ... & Wu, X. (2025). _Seeing helps hearing: A multi-modal dataset and a mamba-based dual branch parallel network for auditory attention decoding._ _Information Fusion_, 118, 102946.
>
> [2] Jiang, Y., Chen, N., & Jin, J. (2022). _Detecting the locus of auditory attention based on the spectro-spatial-temporal analysis of EEG._ _Journal of Neural Engineering_, 19(5), 056035.

---

> > ### Author Rebuttal · Reviewer_Eoyt · 2026-04-06
> >
> > The authors’ detailed response has effectively addressed the concerns.
> >
> > Table 3 reports LOSO only on DTU. Without cross-dataset LOSO validation, how confident are you in the cross-subject generalization claim... It is still one of the concerns.
> > In Table 3, DBPNet on DTU (LOSO) shows 53.7%, but in Table 1 (within-subject) it shows 94.4%. It is quite difference between intra-, and inter subject... It seems that model advancements is absolutely necessary.
> > This paper makes a great contribution to the community. But some concerns are remains.. therefore I maintain my original rating.

---

> > > ### Author Response · Authors · 2026-04-07
> > >
> > > Thank you very much for your careful and thoughtful follow-up comments. We sincerely appreciate your recognition of our detailed response, and we are especially grateful for your continued attention to the cross-subject generalization issue.
> > >
> > > We fully agree that LOSO evaluation is substantially more challenging than the within-subject setting, and that the large gap between intra-subject and inter-subject results reflects a fundamental difficulty of EEG-based auditory attention decoding rather than a dataset-specific phenomenon. In the cross-subject setting, inter-subject variability in EEG distributions, anatomical differences, cognitive strategies, and recording noise can all significantly weaken the transferability of subject-specific discriminative patterns. Therefore, we agree that reporting LOSO results on only one dataset is not sufficient to support an overly strong cross-subject generalization claim.
> > >
> > > Our intention in adding the DTU LOSO experiment was to provide preliminary evidence under a more challenging and realistic setting, rather than to suggest that cross-subject generalization has been fully resolved. We selected DTU because it presents relatively complex auditory conditions and thus serves as a meaningful first testbed for robustness beyond the within-subject scenario. Although the absolute LOSO accuracies are much lower than the within-subject results for all methods, PCRNet still achieves the best performance among the compared baselines under this difficult setting. We will revise the manuscript to make this point clearer and further soften the corresponding claim, emphasizing that our current contribution mainly lies in improving decoding performance in the within-subject setting, while the LOSO result should be interpreted as initial evidence of robustness rather than a definitive solution to cross-subject generalization.
> > >
> > > We also agree with your view that further methodological advances are necessary for this problem. In future work, we plan to extend the cross-subject evaluation to additional datasets and investigate subject-invariant modeling strategies, such as domain adaptation, distribution alignment, and more robust cross-subject representation learning, to better reduce the gap between intra-subject and inter-subject decoding performance.
> > >
> > > Thank you again for your valuable comments and for helping us clarify the scope and limitations of our work.

---

### Official Review · Reviewer_JfA7 · 2026-03-13

**Soundness:** 3
**Presentation:** 3
**Significance:** 3
**Originality:** 3
**Overall Recommendation:** 5
**Confidence:** 5

**Summary:**

This paper introduces PCRNet, a novel framework for EEG-based Auditory Attention Decoding (AAD). The core contribution lies in preserving and refining phase information in the frequency domain through a Dual-Domain Integration (DDI) module. The authors report state-of-the-art results on the KUL, DTU, and AVED datasets.

**Compliance With Llm Reviewing Policy:**

Affirmed.

**Final Justification:**

Referring to the Rebuttal Acknowledgement

**Key Questions For Authors:**

1. The main weaknesses of this work are outlined above. The authors are expected to provide thorough clarifications regarding these concerns.

2. What are the specific differences between audio-only (AO) and audio-visual (AV) scenes in the AVED dataset, and what impacts do these differences have on the experimental results of the proposed model?

3. In Equation (5), the self-attention calculation uses both a learnable temperature parameter and a fixed scaling factor. What is the motivation for using both? Does the learnable temperature significantly improve the quality of the attention map compared with using only the standard scaling factor?

4. In the Temporal Context Calibration (TCC) module, specific kernel sizes are adopted for the Multi-scale Temporal Attention. Why were these particular scales chosen? How do they relate to the intrinsic EEG frequency bands?

5. Does the number of EEG channels and the sampling rate affect the AAD accuracy of the proposed PCRNet?

**Limitations:**

Yes

**Strengths And Weaknesses:**

Strengths:

1. The proposed method is theoretically well motivated and highly convincing from the perspective of phase dynamics. By pointing out that traditional AAD methods often neglect temporal alignment information embedded in phase, the paper establishes a strong and novel rationale for the proposed approach.

2. Architecturally, the method represents a meaningful advancement over conventional time-frequency networks. The model explicitly captures phase information through complex-valued processing, making the overall design both elegant and technically rigorous.

3. The submission is very thorough, with a highly informative Appendix. The extensive supplementary materials, especially the visual decompositions of EEG signals and the complete algorithm pseudocode, substantially clarify the authors’ motivation and also improve the reproducibility of the work.

Weaknesses:

1. Although the authors mention cross-subject analysis as future work in the Conclusion, an important methodological discussion is still missing. Since phase information in EEG is highly sensitive to individual differences and subtle electrode placement variations, relying on fine-grained phase features may further aggravate the challenge of cross-subject generalization compared with magnitude-only approaches. I suggest adding a more explicit discussion on this point.

2. There are some writing issues in the manuscript. For example, in the heading of Section 2.3 (Lines 185–186), “Residual” is mistakenly written as “Resisual”.

---

> ### Author Rebuttal · Authors · 2026-03-30
>
> We sincerely thank you for your insightful feedback. Below, we address each comment individually.
>
> ---
>
> ### 1. Weaknesses of the work
> Thank you very much for this valuable suggestion. We agree that cross-subject generalization is an important limitation and deserves a clearer discussion. In the revised Discussion, we have added a dedicated paragraph to clarify this trade-off more explicitly. We would like to emphasize that this limitation does not imply that phase information should be discarded; rather, it suggests that future work should incorporate better alignment strategies to preserve the benefits of phase-aware modeling while improving cross-subject generalization.
>
> We also thank the reviewer for catching the typo in Section 2.3. We will correct it in the revised manuscript and carefully proofread the paper again to eliminate similar issues.
>
> ---
>
> ### 2. Differences between AO and AV scenes and their impacts
> Thank you for raising this question. In Audio-Only (AO) scenes, subjects rely entirely on auditory input to attend to the target speaker. In Audio-Visual (AV) scenes, subjects are additionally provided with visual cues—such as the speaker’s lip movements and facial expressions—that facilitate multisensory integration and strengthen attentional tracking.
>
> Our experimental results are consistent with this interpretation. The proposed model generally achieves higher or more stable performance in AV scenes, which we believe is due to the stronger and more robust neural tracking patterns induced by multimodal stimulation.
>
> ---
>
> ### 3. Motivation for both learnable temperature and fixed scaling in Eq. (5)
> Thank you for raising this question. The fixed scaling factor $ \sqrt{d_k}  $ follows standard attention practice and is used to stabilize the magnitude of dot-product attention with respect to feature dimensionality. Without this normalization, the attention logits may become excessively large, making optimization less stable.
>
> The learnable temperature $ \tau  $ serves a different purpose: it provides additional flexibility by allowing each attention head to adaptively control the sharpness of the attention distribution beyond the standard scaling. Therefore, these two terms are complementary rather than redundant— $ \sqrt{d_k} $ provides stable baseline normalization, while $ \tau $ enables adaptive modulation of attention sharpness.
>
> ---
>
> ### 4. Choice of kernel sizes in the TCC module
> Thank you for raising this question. The multi-scale kernels were chosen to capture temporal variations at different resolutions. Their purpose is to provide complementary receptive fields for modeling short-, medium-, and relatively longer-range temporal dependencies that may be useful for auditory attention decoding.
>
> We would also like to clarify that these kernel sizes are not intended to correspond one-to-one with specific EEG frequency bands. Instead, they function as a practical multi-scale temporal modeling strategy for capturing diverse neural dynamics relevant to attention-related patterns.
>
> ---
>
> ### 5. Effect of EEG channels and sampling rate
> Thank you for raising this question. Yes, both the number of EEG channels and the sampling rate can affect AAD accuracy. In general, a larger number of channels provides richer spatial information, while a higher sampling rate preserves more detailed temporal dynamics. Together, these factors improve the quality of the spatiotemporal representations available to the model.
>
> For example, the KUL dataset—with 64 channels and a sampling rate of 128 Hz—provides richer spatiotemporal information than datasets with lower spatial or temporal resolution, which likely contributes to its stronger performance.

---

> > ### Author Rebuttal · Reviewer_JfA7 · 2026-04-02
> >
> > The authors’ detailed response has effectively addressed the concerns.
> > The authors provide a comprehensive discussion of cross-subject generalization and further clarify the direction of future work. The interpretation of AO and AV scenes, the motivation for introducing a learnable temperature parameter and a fixed scaling factor, and the choice of kernel sizes in TCC are technically sound and improve the overall experimental design. The clarification regarding the effects of EEG channels and sampling rate also helps explain the performance differences across datasets.
> >
> > Overall, the paper’s main claims are well supported, and it is considered a strong and valuable contribution. Therefore, the rating is increased.

---

> > > ### Author Response · Authors · 2026-04-02
> > >
> > > We sincerely thank you for your positive feedback and for raising the rating of our paper. We greatly appreciate your recognition of the overall contribution of our work.

---

### Decision · Program_Chairs · 2026-04-30

**Decision:**

Accept (regular)

**Comment:**

The paper proposes a novel model for auditory attention decoding using EEG to identify the attended speaker in a multi-speaker setting by utilizing the oft-overlooked phase information. PCRNet is a multi-module system with modules dedicated to capture long-range temporal dependencies through multi-scale temporal attention, and phase-guided spectral filtering. The paper has received 4 reviews with three reviewers inclined to accept the paper (5,5,6) and one reviewer recommending a reject (3).

Most reviewers found the paper to be theoretically well motivated and with a high biological plausibility. The design principle of including the oft-overlooked phase information is reasonable. The architecture is seen to be elegant and technically sound, and a meaningful advance over conventional T-F networks. The evaluation to be broad and the experimental results extensive. The SOTA accuracy under the challenging ultra-short 0.1-second decision windows is appreciated and seen as a significant contribution toward low-latency BCI applications.

On the other hand, some reviewers see this as a systems paper with known modules rather than principled methodological novelty. Reviews and author responses reveal the need for more ablations to establish the benefit of individual modules and statistical significance testing to establish the benefit of using phase information (spreads are quite high). There are pending concerns whether the topography maps effectively support the conclusion that the method focuses on neurophysiologically relevant areas and the need to decouple performance gains from the application of CSP.

All in all, while the paper utilizes oft-looked phase information and seems biologically plausible, with promising performance in the ultra-short 0.1 second decision window, it needs more work to establish that the gains are statistically significant, neurophysiologically relevant, and are well-grounded in the design choices.